# *Cyberlindnera fabianii*, an Uncommon Yeast Responsible for Gluten Bread Spoilage

**DOI:** 10.3390/foods13152381

**Published:** 2024-07-27

**Authors:** Andrea Colautti, Elisabetta Orecchia, Francesca Coppola, Lucilla Iacumin, Giuseppe Comi

**Affiliations:** 1Department of Agricultural, Food, Environmental and Animal Science, University of Udine, Via Sondrio 2/a, 33100 Udine, Italy; andrea.colautti@uniud.it (A.C.); orecchia.elisabetta@spes.uniud.it (E.O.); lucilla.iacumin@uniud.it (L.I.); 2Food Sciences Institute, National Research Council, Via Roma, 64, 83100 Avellino, Italy; fracop93@gmail.com

**Keywords:** gluten bread, *Cyberlindnera fabianii*, spoilage, volatile compounds

## Abstract

A single strain of yeast was isolated from industrial gluten bread (GB) purchased from a local supermarket. This strain is responsible for spoilage consisting of white powdery and filamentous colonies due to the fragmentation of hyphae into short lengths (dust-type spots), similar to the spoilage produced by chalk yeasts such as *Hyphopichia burtonii*, *Wickerhamomyces anomalus* and *Saccharomycopsis fibuligera*. The isolated strains were identified initially by traditional methods as *Wickerhamomyces anomalus*, but with genomic analysis, they were definitively identified as *Cyberlindnera fabianii*, a rare ascomycetous opportunistic yeast species with low virulence attributes, uncommonly implicated in bread spoilage. However, these results demonstrate that this strain is phenotypically similar to *Wi. anomalus*. *Cy. fabianii* grew in GB because of its physicochemical characteristics which included pH 5.34, Aw 0.97 and a moisture of about 50.36. This spoilage was also confirmed by the presence of various compounds typical of yeasts, derived from sugar fermentation and amino acid degradation. These compounds included alcohols (ethanol, 1-propanol, isobutyl alcohol, isoamyl alcohol and n-amyl alcohol), organic acids (acetic and pentanoic acids) and esters (Ethylacetate, n-propil acetate, Ethylbutirrate, Isoamylacetate and Ethylpentanoate), identified in higher concentrations in the spoiled samples than in the unspoiled samples. The concentration of acetic acid was lower only in the spoiled samples, but this effect may be due to the consumption of this compound to produce acetate esters, which predominate in the spoiled samples.

## 1. Introduction

Italian bread consumption has decreased over the years, both because of a loss of interest on the part of consumers, who tend to prefer foods based on fresh vegetables and fruit, and because of the decrease in the number of artisan bakeries. In fact, 84.9% of Italians have not given up fresh bread and continue to purchase it from the 20 thousand artisan bakeries which churn out approximately 1.5 million tonnes a year [1]. However, they have bought less than in the past and less frequently, so much so that over 40 years, consumption has decreased by 65%. At the same time, the sales of industrial breads have exploded, reaching over 216 tonnes, as well as those of other products such as sandwich bread (+8.5% in volume), hamburger buns (+8.3%) and pancarrè (+5%). A trend toward bread produced with sourdough can be confirmed, being the best-selling product after common bread [2]. Consumers also like special recipes, such as multigrain and healthy breads, and small formats and loaves are preferred because they last for several days. The long shelf-life of bread is crucial to its purchase, especially in terms of its anti-waste properties [2]. For this reason, over the years, the industry has churned out quality breads, whole or presliced, packaged them in a modified atmosphere and supplemented them with additives. In particular, the shelf-life of industrial breads has increased because of the use of modified atmospheres of 100% CO_2_ or the addition of ethanol in packages in which the atmosphere is represented by air [3]. For months, trade associations have been asking for interventions to combat high costs and save artisanal bread without having to increase prices, which have risen by 57% in 10 years [4]. Finally, the decrease in requests for gluten bread (GB) is motivated by the increased prevalence of celiac disease, as well as a growing population turning to healthier diets, leading to a growing demand for gluten-free products (GFPs), especially bakery products [5,6]. Owing to dietary restrictions, people with celiac disease constitute a special group of consumers with specific attitudes and needs, and they drive the gluten-free bread (GFB) industry [7,8,9]. In recent years, the sensorial and nutritional qualities of GFB and other bakery goods such as cookies, muffins or sponge cakes have reached high levels and are able to satisfy consumers with celiac disease. This is demonstrated by the high demand for GFB products and by various scientific papers concerning GFB production and technology [10,11].

Italian supermarkets sell different types of breads produced by local industries or that are imported. Their quality is highly dependent on the brand, the technological process and the flours used.

However, it is possible that during storage at room temperature, yeasts and molds can grow, producing spoilage consisting of slime and off-odors. The spoilage of GB is a major concern in the food industry and leads to considerable food loss [12]. Major losses occur in industrial GB, e.g., toast bread and modified atmosphere packaged (parbaked) bread, which is stored for a longer period than traditional fresh bread [12,13]. Yeasts and molds represent the main causes of spoilage of bread worldwide [14,15]. Various species of the genera *Penicillium* spp. or *Aspergillus* spp. [16] and chalk yeasts, also called chalk molds although they are yeasts [17], are characterized by the ability to produce white powdery and filamentous colonies due to the fragmentation of hyphae into short lengths (dust-type spots) and are the main spoilers of industrial bread [14,17,18,19,20]. In particular, chalk yeasts, resembling molds, are the most common on sliced and rye breads.

The high moisture and nutrient richness of bread allow for yeast and mold growth [14,17,20], which results in spoilage due to several defects including visible molding, off-flavors and odors, thus causing significant economic losses for the bakery industry [13]. Among chalk yeasts, the dominant species include *Saccharomycopsis fibuligera* [21], *Hyphopichia burtonii* [17,21], *Zygosaccharomyces bailli* and *Saccharomyces cerevisiae* [18]. In addition, *Wickerhamomyces anomalus* (formerly known as *Pichia anomala*) is responsible for the spoilage of bread but does not belong to the chalk yeasts, even though some authors consider it to be part of that group [19]. Indeed, usually, either *Sa. fibuligera* and *H. burtonii* or *Wi. anomalus* rather than *Hansenula anomala* and *Candida pelliculosa*, which produces white pseudomycelium and hypha-like structures, are responsible for the spread of white and powdery colonies that look like sprinkled chalk dust on the surface of the product. In particular, the chalky aspect of *S. fibuligera* and *H. burtonii* is due to their typical growth structures resembling hyphae and/or pseudomycelium consisting of chains of budded yeast cells that did not separate after duplication [17]. The type of packaging and the addition of ethanol as a preservative do not stop chalk yeast growth. Indeed, *Wi. anomalus*, *Sa. fibuligera* and *H. burtonii* spoil modified atmosphere packaging (MAP, 100% CO_2_) parbaked breads [17] but are also frequently isolated from spoiled industrial bread packaged in air [20]. In particular, *Wi. anomalus* is the main parent of Belgian parbaked bread packaged under a modified atmosphere [17]. The spoilage process also includes the production of high amounts of ethyl acetate from glucose or ethanol at Aw values higher than 0.87. *Wi. anomalus* is primarily responsible for ethyl acetate production, which confers an undesirable smell, is identified as a “chemical odor” and produces consumer complaints [19].

The origin of the contamination is not well known, but surely, it occurs after baking considering the cooking temperature, which can reach over 180 °C and can kill either molds or yeasts. According to Giannone et al. [20], fungal cells are carried by bioaerosols within bakery plants. They then grow on bread during storage either at room temperature or at refrigerated temperatures and regardless of the type of packaging (air, vacuum or MAP) or preservative added [22].

Recently, the presence of spots of white yeasts on GB packaged in MAP (100% CO_2_) was observed. These spots constituted one lot and were stored at ambient temperature. The spots were not similar to those produced by chalk yeasts but resembled a floury patina. In addition, after opening the package, a vinegary aroma was perceived. Therefore, the aim of this work was to determine the microbial agent responsible for spoilage, which appears to differ from that produced by chalk yeasts.

## 2. Materials and Methods

### 2.1. Microbial Analysis

The sliced GBs were made with 0 wheat flour (water 13%; Protein 11%; Lipids 1%; carbohydrates 74%; Ash 0.65%), water, vegetable fiber, dextrose, modified cellulose, yeast, salt, tartaric acid and citric acid and packaged in a modified atmosphere (100% CO_2_) without the addition of ethanol as an antimicrobial agent. The GB was foiled after 6 h from cooking and kept at room temperature. On day 15, the white spots were already visible. The package film was made with a high barrier grade EVOH with a density of 1.21 g/cm^3^, a melting point of 190 °C and an oxygen transmission rate of 0.1 (cm^3^ × 20 μm)/(m^2^ × day × atm). The superior property of EVOH is that it generally has extremely low permeability at low relative humidity (RH). However, under high moisture conditions, gases can permeate easily.

The microbial spots were present in 5 out of the 10 GB packets, all of which belonged to the same lot (22 January 2024). All the spots of each spoiled packet were analyzed via the traditional method. Briefly, the area of each spot was collected and diluted in sterile peptone water (NaCl, 6 g; peptone, 1 g; distilled water, 1 L). Then, 0.1 mL of each dilution was plated on malt agar (MA, Oxoid, Milan, Italy) and incubated at 25 °C for 3–5 days. The growth colonies were counted. From the MA plates, which contained 30 to 300 colonies, 100 colonies were randomly isolated. These were selected regardless of morphology, color or size. The isolated colonies were purified on MA agar and then stored at −80 °C in MRS broth supplemented with glycerol (30% Sigma–Aldrich, Schnelldorf, Germany).

### 2.2. The Identifications of the Isolated Yeasts

The 100 isolated colonies were identified using traditional methods as reported by Kurtzman et al. [23] and Kurtzman [24]. Briefly, growth on 5% MA and malt broth (MB, Oxoid, Italy), growth on morphology agar (MA with cover glass) and ascospore formation and fermentation and assimilation tests were performed. API 50 CH (BioMérieux, Florence, Italy) was used for fermentation and assimilation tests. Yeast nitrogen broth (Oxoid, Milan, Italy) and yeast extract broth (Oxoid, Milan, Italy—0.5% yeast extract and 1000 mL distilled water) were used for the assimilation and fermentation tests, respectively.

The mycelium and cell morphology and the presence of spores were observed under an optical microscope.

Considering that the traditional test could not be updated and provides subjective results depending on the analyst, additional tests were performed. In particular, ten strains were subjected to the following test. For identification, after the DNA was extracted via the phenol–chloroform method [25], a ~600 bp portion of the D1–D2 region of the large-subunit rRNA gene was sequenced using primers NL1 (5′-GCCATATCAATAAGCGGAAAAG-3′) and NL4 (5′-GGTCCGTGTTTCAAGACGG-3′). The reaction mixture included 1X PCR Buffer I, 0.2 mM each dNTP, 0.2 µM each primer, 100 ng template DNA and 1.25 U AmpliTaq DNA Polymerase (Applied Biosystem, Milan, Italy) at a final volume of 50 µL with sterile water. The cycling conditions were as follows: 30 cycles of 95 °C for 60 s, 48 °C for 45 s and 72 °C for 60 s, with initial denaturation at 95 °C for 5 min and a final extension at 72 °C for 7 min. After purification, the PCR products were sequenced by Eurofins Genomics (Ebersberg bei München, Germany). The sequences were aligned with those in GenBank using the BLAST suite [26].

The sequences were then aligned with the reference genomes from NCBI for *Candida*/*Cyberlindnera* spp. and *Wickerhamomyces anomalus* to obtain the corresponding sequences. These sequences were compared to construct a phylogenetic tree using MAFFT [27] and FastTree [28] and then plotted with FigTree [29].

### 2.3. Physicochemical Determination

The pH of GB was measured directly by inserting a pH meter (Radiometer, København, Denmark) into the sample. The water activity (Aw) was determined using a Hygromer AWVC (Rotronic, Milan, Italy). The final values of all the above physicochemical parameters were expressed as the average of the measurements of ten samples. The moisture content was determined according to A.O.A.C. [30]. The modified atmosphere was determined using a Gas Analyzer for MAP packages (OXYBABY M+, standard version, WITT-GASETECHNIK GmbH & Co KG, Witten, Germany).

### 2.4. Volatile Compound Analysis

Volatile compounds were identified using SPME-GC–MS on a Finnigan Trace DSQ (Thermo Scientific Corporation, Waltham, MA, USA) with an Rtx-Wax capillary column (length 30 m × 0.25 mm id., film thickness 0.25 µm; Restek Corporation, Waltham, MA, USA) according to the method reported in Chiesa et al. (2006) [31]. SPME sampling was performed by exposing divinylbenzene/carboxen/polydimethylsiloxane fibers (50/30 μm, 2 cm long from Supelco Ltd., Bellefonte, PA, USA) for 30 min in the headspace of the spoiled and unspoiled packages before opening.

The volatile compounds were identified by comparing the spectra obtained with the spectra available in the commercial Wiley library and from an internal library. The results are expressed as the average of 5 spoiled and 5 unspoiled samples analyzed in triplicate.

### 2.5. Statistical Analysis

The data were analyzed using Statistica 7.0 version 8 software (StatSoft Inc., Tulsa, OK, USA, 2008). The values of the different parameters were compared with a one-way analysis of variance, and the means were then compared using Tukey’s honest significance test. Differences were considered significant at *p* < 0.05. Each physical–chemical and microbial analysis included 10 samples of either spoiled or unspoiled goose sausages. Three samples were tested for volatilome analysis.

## 3. Results and Discussion

### 3.1. The Identification of the Strains

The results of the analysis revealed that the concentration of the yeasts at each spot was approximately 8–9 CFU/cm^2^. From the plates containing approximately 30–300 CFUs, 100 colonies were randomly selected and subjected to identification. The traditional test identified all the strains tested as *Wickerhamomyces anomalus* (formerly known as *Pichia anomala*), a nonchalk yeast responsible for bread spoilage [12]. However, this yeast is often considered part of the group of chalk yeasts by other authors [19] because they tend to grow structures resembling hyphae and form mycelia. In particular, it forms pseudohyphae, chains of budded yeast cells that do not separate after duplication.

Indeed, the phenotypic results demonstrated the following:(a)After 3 days of growth on 5% MA at 25 °C, the cells were spherical to ellipsoidal, 3.3–5 × 3.9–9.5 μm and occurred singly, in pairs or in small clusters.(b)The colonies were butyrous and white in color.(c)Under a light microscope, careful examination revealed rarely long, slender crystals among the cells.(d)In malt broth (MB, Oxoid, Milan, Italy), the strains produced thin, smooth and waxy pellicols.(e)When MA was covered with glass, the strains produced abundant and highly branched pseudohyphae after 7 days at 25 °C.(f)All the strains also produced real mycelia with true hyphae.(g)In MA, the colonies appeared white, faintly glistening to dull and butyrous, with lobed and fringed margins, due to pseudohyphae.(h)An ester-like odor was often present.

Conversely, the genomic study identified all the strains as *Cyberlindnera fabianii*. The strains were named PP883794. The obtained sequence, with a length of 571 bp, was deposited in NCBI under the accession number PP883794. Indeed, the initial alignment using BLASTn revealed that the sequence had the highest match, with 16 sequences identified as *Cy. fabianii*, showing 568 identities and 2 gaps (MN054504.1, MK394133.1, MH472646.1, PP033888.1, KY108793.1, KY108792.1, KY107357.1, KY107356.1, KY107353.1, KU170644.1, CP048731.1, JQ342084.1, MT860205.1, NG_055731.1, AM397861.1, LK392383.1).

For a more accurate analysis using verified sequences, the sequence was aligned with the reference genomes of the genus *Candida*/Cyberlindnera. As shown in Table 1, *Cy. Fabianii* strain JOY008 presented the highest percentage identity (99.47%), followed by *Cy. veronae* (98.60%). 

To gain a deeper understanding of the relationships between these different species, a phylogenetic tree was plotted using sequences aligned with MAFFT. This unrooted tree (Figure 1) revealed that the analyzed strain was placed on the same branch as the reference strain of *Cy. fabianii*, in close proximity. The closest species grouped in the same clade were *Cy. veronae*, *Cy. americana*, *Cy. mycetangii* and *Cy. maritima*, while *Cy. mississipiensis* and *Cy. amylophila* were more distantly related and grouped together on a different branch, followed by *Cy. xishuangbannaensis*. In contrast, the reference strains of *Wi. anomalus*, a species reported in the literature to have morphological traits that can be confused with *Cy. Fabianii* [32], was placed in a very distant clade which also included the reference strains of *Cy. galapagoensis* and *Cy. samutprakarnensis*. Given the significant differences reported in Table 1, it was possible to exclude this strain from belonging to the species *Wi. anomalus*.

According to the results of the genomic study, the isolated strains were identified as *Cyberlindnera fabianii*. The nomenclature of *Cy. fabianii*, an ascomycetous yeast, has changed several times [33]. It was formerly referred to as *Cy. fabianii* [34] or *Hansenula fabianii* or *Pichia fabianii* or *Lindnera fabianii* [35]. The members of the genus *Cyberlindnera* are widely distributed in the environment, particularly as contaminants in food, fermentation products and industrial waste. In particular, it has been used for biotechnological procedures, for the treatment of wastewater from food processing plants and for the bioremediation of soils contaminated with organic and inorganic compounds [36]. It can be found in environments such as alcoholic beverages, sugarcane and exotic fruits [35,37].

Indeed, in recent years, it has been studied for industrial uses. For enological fermentation, Vicente et al. [38] studied the growth fitness and fermentative potential and enological impact of three wine yeast strains belonging to three nonconventional species, Cy. fabianii (formerly *Pichia fabianii*), *Kazachstania unispora* (formerly *Saccharomyces unisporus*) and *Naganishia globosa* (formerly *Cryptococcus saitoi*), and investigated several ecological aspects (niche preferences and competitive capacity) to understand their interaction patterns with S. cerevisiae [38].

In the bioremediation field, researchers have studied environmentally friendly, economically viable and socially acceptable techniques to remove contaminants from the environment [39,40,41]. In particular, many microorganism enzymes are isolated, purified or partially purified to catalyze the detoxification of contaminants, and these techniques are becoming important because [42] free enzyme bioremediation is not dependent upon the growth of intact organisms; hence, the rate of detoxification is directly linked to the catalytic properties and concentration of applied enzymes [43]. Recently, free and immobilized laccases were characterized from *Cy. fabianii* to apply in the degradation of bisphenol A [44].

Finally, *Cy. fabianii* is an uncommon and rare ascomycetitive opportunistic yeast species with low virulence attributes, despite some authors considering it a medically important fungus [45,46,47], having been shown to cause invasive bloodstream infections [32,34,48,49].

### 3.2. Physicochemical Parameters of Gluten Bread

Table 2 shows the physicochemical parameters of spoiled and unspoiled GB. The data show no significant differences between the investigated samples (*p* > 0.05). In addition, all the parameters show that GB is a useful substrate for yeast growth, considering that it is stored at ambient temperatures (20–25 °C). Burgain et al. (2015) [19] previously described the growth of xerophilic and chalk yeasts, including *Hyphopichia burtonii*, *Wickerhamomyces anomalus* and *Saccharomycopsis fibuligera*, in bread and demonstrated that they could grow at 0.85 Aw at 15–25 °C and pH 4.6–6.8.

Recently, Debonne et al. [12] also assessed the growth of *Hy. burtonii*, *Wi. anomalus* and *Sa. fibuligera* in commercial bread through multiple methods, including radial growth, biocontrol assays, growth/no-growth modeling and bread challenge tests, the influence of pH (4.5–6.5), temperature (22–30 °C), propionic acid (0–0.3%) and the presence of other spoilage fungi. These authors demonstrated that pH and temperature do not play major roles in the growth of all the investigated strains. Only *Wi. anomalus* showed optimal growth at pH 4.5 and 30 °C on malt extract agar.

Considering that the strains isolated from *Cy. fabianii* have phenotypic characteristics similar to those of chalk yeasts, the combined effects of temperature, Aw, pH and moisture of the GFB cannot prevent its growth.

### 3.3. Identification of Volatile Compounds in Spoilage

A study of volatile compounds confirmed the activity of *Cy. fabianii* in spoiled GB. Indeed, the levels of some alcohols, carboxylic acids and esters increased in the spoiled GB samples. This feature is emphasized in Table 3, which lists only the components whose concentrations varied considerably between the spoiled and the unspoiled GB (*p* < 0.05).

Since the concentrations of aldehydes, ketones and hydrocarbons did not differ between the spoiled and unspoiled samples, they were not reported. Indeed, the concentrations of 3 ketones, 8 aldehydes and 15 hydrocarbons did not change in either the spoiled or unspoiled GB (*p* > 0.05). Conversely, the amount of ethanol in the spoiled samples was greater than that in the unspoiled GB samples, reaching a significant difference (*p* < 0.05). In addition, the concentrations of higher alcohols were different among the GB samples. In particular, their concentration was greater in the spoiled samples (*p* < 0.05). Yeasts produce higher alcohols directly from sugar fermentation or by amino acid degradation [50,51,52,53]. The higher alcohols detected included isobutyl alcohol, isoamyl alcohol and 1-propanol, which could originate from valine, leucine and threonine, respectively [52,54]. Similarly, in the spoiled samples, the acetic acid concentration was noticeably lower, but it could be hypothesized that it was used by yeasts to produce acetic esters [52]. Indeed, the concentrations of both ethyl and acetic esters were greater in the spoiled samples (*p* < 0.05). In particular, as shown in Table 3, the identified esters included ethyl acetate, n-propylacetate, ethyl butyrate, isoamyl acetate and ethyl pentanoate. The esters were not identified in the unspoiled samples except for isoamyl acetate, but its concentration was low, demonstrating that in the spoiled samples, they were produced by *Cy. fabianii*. Esters are formed via an intracellular process catalyzed by an acyl transferase or ‘ester synthase’ [55,56]. The reaction requires energy provided by the thioester linkage of the acyl-CoA cosubstrate. The most abundant acyl-CoA is acetyl-CoA, which can be formed either by the oxidative decarboxylation of pyruvate or by the direct activation of acetate with ATP [52,53,54,55,56]. The majority of acetyl-CoA is formed by the oxidative decarboxylation of pyruvate, while most of the other acyl-CoAs are generated by the acylation of free CoA catalyzed by acyl-CoA synthase (fatty acid metabolism) [52]. Esters are considered flavor-active compounds. Consequently, their production is necessary for the aroma of fermented beverages such as wine or beer [51,52]. These aromas include two groups: The first is represented by acetate esters (the acid group is acetate; the alcohol group is ethanol or a complex alcohol derived from amino acid metabolism) such as ethyl acetate (solvent-like aroma), isobutyl acetate (fruity aroma), phenyl ethyl acetate (roses, honey) [52] and isoamyl acetate (banana aroma), which is the most influential acetate ester in most beer, (white) wines and sake [51]. The second group comprises ethyl esters (the alcohol group is ethanol, and the acid group is a medium-chain fatty acid), which includes ethyl hexanoate (aniseed, apple-like aroma) and ethyl octanoate (sour apple aroma). Among these two groups, acetate esters have received the most past attention, not because they are more important but because they are produced at much higher levels and are therefore easier to measure [52].

Finally, all the volatile compounds identified in the spoiled samples do not influence the healthiness of the bread, being present in many foods and fermented products such as wine [50,51,52,53,54]. So, the higher concentration of the volatile compounds in the spoiled GB compared to unspoiled only confirms the activity of *Cy. fabianii.* The spoilage observed is due to the presence of white spots and the increase in some volatile compounds’ concentration.

## 4. Conclusions

An uncommon yeast identified as *Cy. fabianii* was isolated from spoiled GB. This strain is phenotypically similar to *Wy. anomalus*. Indeed, using traditional methods, it was previously identified as *Wi. anomalus*. However, applying genomic methods, the isolated strains were clearly identified as *Cy. fabianii*. This strain produced the spoilage of GB, resulting in white powdery and filamentous colonies due to the fragmentation of hyphae into short lengths (dust-type spots). Considering the type of spoilage produced by *Hy. burtonii*, *Wi. anomalus* and *Sa. fibuligera*, *Cy. fabianii* can be considered a chalk yeast and is typically responsible for bread spoilage. In addition, spoilage was confirmed by the presence of different compounds derived from sugar fermentation and amino acid degradation.

In the spoiled GB samples, different alcohols, organic acids and esters were identified at higher concentrations than in the unspoiled samples. The concentration of acetic acid was lower only in the spoiled samples, but this effect may be due to the use of this compound to produce acetate esters, which predominate in the spoiled samples.

## Figures and Tables

**Figure 1 foods-13-02381-f001:**
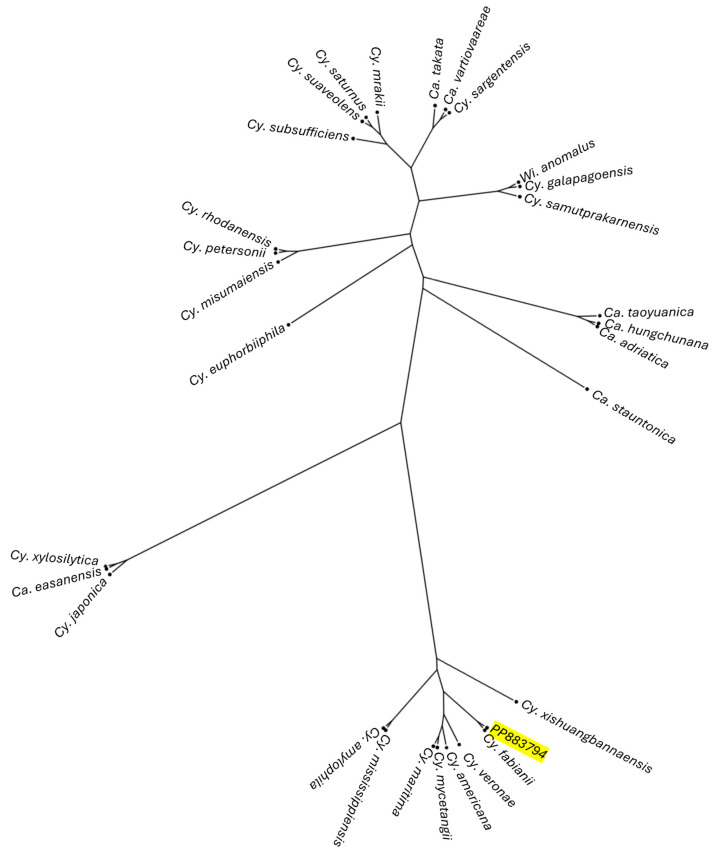
Phylogenetic tree of reference strains of *Candida*/*Cyberlindnera* spp. and *Wi. anomalus*.

**Table 1 foods-13-02381-t001:** Reference genomes used for BLAST comparison, ordered for decreasing identity.

Species	Strain	Genome	BLAST Comparison with PP883704		
			Length	Identities	Gaps
*Cyberlindnera fabianii*	JOY008	ASM2264183v1	571	568	2
*Cyberlindnera veronae*	NRRL Y-7818	ASM3056306v1	571	563	2
*Candida stauntonica*	CBS 12241	ASM3055833v1	571	562	2
*Cyberlindnera americana*	NRRL Y-2156	ASM370879v3	571	561	2
*Cyberlindnera mississippiensis*	NRRL YB-1294	ASM3058171v1	571	561	2
*Cyberlindnera xishuangbannaensis*	CBS 14692	ASM3058508v1	571	560	2
*Cyberlindnera amylophila*	NRRL YB-1287	ASM3055715v1	573	559	4
*Cyberlindnera xylosilytica*	NRRL YB-2097	ASM370828v2	571	557	2
*Cyberlindnera japonica*	NRRL YB-2750	ASM3055717v1	571	556	2
*Cyberlindnera mycetangii*	NRRL Y-6843	ASM370829v3	571	556	2
*Candida easanensis*	JCM 12476	ASM3058013v1	571	556	2
*Candida taoyuanica*	CBS 12242	ASM3056705v1	571	555	2
*Cyberlindnera maritima*	NRRL Y-17775	ASM3055734v1	571	552	2
*Cyberlindnera subsufficiens*	NG8.2	ASM1794857v1	575	548	6
*Candida hungchunana*	CBS 12243	ASM3056322v1	571	548	2
*Candida adriatica*	CBS 12504	ASM3055813v1	571	547	2
*Cyberlindnera euphorbiiphila*	NRRL Y-12742	ASM3055662v1	573	546	4
*Cyberlindnera mrakii*	NRRL Y-1364	ASM370644v3	575	546	6
*Cyberlindnera saturnus*	NRRL Y-17396	ASM370924v3	575	545	6
*Cyberlindnera suaveolens*	NRRL Y-17391	ASM370822v3	575	545	6
*Cyberlindnera sargentensis*	SHA 17.2	ASM2099542v1	576	544	7
*Candida vartiovaarae*	DDNA#1	CvDDNA1.1	575	544	6
*Candida takata*	CBS 12244	ASM3056414v1	575	543	6
*Cyberlindnera misumaiensis*	NRRL Y-17389	ASM370774v2	577	530	10
*Cyberlindnera samutprakarnensis*	CBS 12528	ASM3056971v1	585	530	16
*Cyberlindnera rhodanensis*	NRRL Y-7854	ASM3056939v1	576	529	9
*Cyberlindnera galapagoensis*	CBS 13997	ASM3056520v1	577	524	10
*Cyberlindnera petersonii*	NRRL YB-3808	ASM3057413v1	568	521	5
*Wickerhamomyces anomalus*	NRRL Y-366-8	Wican1	574	515	5

**Table 2 foods-13-02381-t002:** Physicochemical parameters of gluten bread.

Parameters	Gluten Bread	
	Spoiled	Unspoiled
pH	5.34 ± 0.04 a	5.30 ± 0.03 a
Aw	0.978 ± 0.01 a	0.977 ± 0.02 a
Moisture%	50.36 ± 0.27 a	50.38 ± 0.23 a

The means with the same letters following the lines are not significantly different (*p* < 0.05).

**Table 3 foods-13-02381-t003:** Volatile compounds of spoiled and unspoiled GB.

Compound	Spoiled	Unspoiled
Ethanol	35.10 ± 0.81 a	12.44 ± 9.65 b
1-propanol	0.74 ± 0.73 a	0.00 ± 0.00 b
Isobutyl alcohol	1.15 ± 1.12 a	0.19 ± 0.32 b
Isoamyl alcohol	6.82 ± 1.69 a	2.95 ± 1.50 b
n-amyl alcohol	0.24 ± 0.17 a	0.15 ± 0.25 b
**ALCOHOLS ***	**44.05 ± 4.52 a**	**15.73 ± 11.72 b**
Ethylacetate	3.63 ± 6.28 a	0.00 ± 0.00 b
n-propyl acetate	0.13 ± 0.23 a	0.00 ± 0.00 b
Ethyl butyrate	0.52 ± 0.28 a	0.00 ± 0.00 b
Isoamylacetate	0.96 ± 1.42 a	0.16 ± 0.28 b
Ethylpentanoate	0.12 ± 0.21 a	0.00 ± 0.00 b
**ESTERS ***	**5.36 ± 8.42 a**	**0.16 ± 0.28 b**
Acetic acid	1.75 ± 0.98 a	2.31 ± 0.71 b
Pentanoic acid	0.38 ± 0.04 a	0.82 ± 0.72 b
**ACIDS ***	**2.13 ±1.02 a**	**3.13 ± 1.43 b**

Data are expressed as the ratio between the area of each peak and the area of the internal standard (4-methyl, 2-pentanol); * sum of compounds; data represent the means ± standard deviations (SDs) of the total samples. The means with the same letters following the lines are not significantly different (*p* < 0.05).

## Data Availability

The original contributions presented in the study are included in the article, further inquiries can be directed to the corresponding author.

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
