# Peer review of "Cyberlindnera fabianii, an Uncommon Yeast Responsible for Gluten Bread Spoilage"

_foods, 2024, doi:10.3390/foods13152381_

Round 1

Reviewer 1 Report

Comments and Suggestions for Authors

Dear Authors,

This manuscript is dealing with the problem which is very common in our life since bread is foodstuff which is cornerstone of human traditional nutrition especially in Europe. The findings from this study could be applied in practice to prevent spoilage of bread and other bakery products. Please find my suggestions below.

In the abstract is not same style of font in the text.

In the abstract are missing the most important values of results from this manuscript. Insert it in abstract.

Explain abbreviation GB when you introduce it for the first time.

Highlight in the introduction what is negative health effects of yeast which you mentioned that are responsible for spoilage.

In the section 2 is not same size of font throughout section.

In the section 3 is highlighted that volatile compounds are responsible for sensor properties of sample. Did volatile compounds which you mentioned in the section 3 affect on health safety of bread? Highlight it in the section 3.  

Did amount of mentioned compounds in the Table 3 affect on healthy safety of bread when they are present in the amounts which are not affect on sensor properties of bread?

In the paragraph from line 327 to 329 you mentioned that different esters were present in the spoiled and unspoiled bread.How then esters from spoiled samples could be used for determination of flavor in unspoiled?

Did presence of esters which are characteristic for spoiled bread could be used in the routine analysis of bread and indicate spoilage? Highlight it in manuscript.

In the conclusion you mentioned gluten free bread. Results are missing for content of compounds of gluten free bread.

Author Response

Dear referee

Enclosed you will find a copy of our revised Manuscript ID: foods-3118685entitled “Cyberlindnera fabianii, an uncommon yeast responsible for gluten bread spoilage” submitted to Foods

The authors would like to thank the reviewers for their careful reading of the manuscript and the resulting constructive comments and suggestions. Basically, we agree with all the points raised by the reviewers, and wherever possible the manuscript has been modified as recommended. All reviewer comments are in black plain font, whereas our response is described in red plain font.

We have made the changes and corrections based on the reviewer’s suggestions. We evaluated the comments and prepared a point-by-point response to each one of them.

Reviewer 1

This manuscript is dealing with the problem which is very common in our life since bread is foodstuff which is cornerstone of human traditional nutrition especially in Europe. The findings from this study could be applied in practice to prevent spoilage of bread and other bakery products. Please find my suggestions below.

In the abstract is not same style of font in the text.

Answer – Thanks for the suggestion – I correct it

In the abstract are missing the most important values of results from this manuscript. Insert it in abstract.

Answer – Thanks for the suggestion – I add the important part – lines 11-29

A single strain of yeast was isolated from industrial gluten bread (GB) purchased from a local supermarket. This strain is responsible for spoilage consisting of white powdery and filamentous colonies due to the fragmentation of hyphae into short lengths (dust-type spots), similar to the spoilage produced by chalk yeasts such as Hyphopichia burtonii, Wickerhamomyces anomalus and Saccharomycopsis fibuligera. The isolated strains were identified initially by traditional methods as Wickerhamomyces anomalus, but with genomic analysis definitively identified as Cyberlindnera fabianii, a rare ascomycetous opportunistic yeast species with low virulence attributes, uncommonly implicated in bread spoilage. However, these results demonstrate that this strain is phenotypically similar to Wi. anomalus. Cy. fabianii grew in the GB because of its physico-chemical characteristic which included pH 5.34, Aw 0.97 and Moisture of about 50.36. This spoilage was also confirmed by the presence of various compounds typical of yeasts, derived from sugar fermentation and amino acid degradation. These compounds included alcohols (Ethanol, 1-propanol, Isobutyl alcohol, Isoamyl alcohol and n-amyl alcohol), organic acids (acetic and pentanoic acids) , and esters (Ethylacetate, n-propil acetate, Ethylbutirrate, Isoamylacetate and Ethylpentanoate), identified in higher concentrations in the spoiled samples than in the unspoiled samples. The concentration of acetic acid was lower only in the spoiled samples, but this effect may be due to the consumption of this compound to produce acetate esters, which predominate in the spoiled samples.

Explain abbreviation GB when you introduce it for the first time.

Answer – Thanks for the suggestion – I introduce it in the Abstract – Line 11 and in the introduction Line 52.

Highlight in the introduction what is negative health effects of yeast which you mentioned that are responsible for spoilage.

Answer – Thanks – The mentioned yeasts in the introduction did not have any negative effects on Health, they are always found in the foods but when they grow the produce a spoilage as shown by the references.

In the section 2 is not same size of font throughout section.

Answer – Thanks I correct it

In the section 3 is highlighted that volatile compounds are responsible for sensor properties of sample. Did volatile compounds which you mentioned in the section 3 affect on health safety of bread? Highlight it in the section 3.  Did amount of mentioned compounds in the Table 3 affect on healthy safety of bread when they are present in the amounts which are not affect on sensor properties of bread.

In the paragraph from line 327 to 329 you mentioned that different esters were present in the spoiled and unspoiled bread. How then esters from spoiled samples could be used for determination of flavor in unspoiled?

Finally, all the volatile compounds identified in the spoiled samples do not influence the healthiness of the bread, being present in many foods and fermented product such as wine [50-54]. So, the higher concentration of the volatile compounds in the spoiled GB respect to unspoiled only confirm the activity of the Cy. fabianii. The spoilage observed is due on the presence of white spots and on the increasing of some volatile compounds concentration.   

Did presence of esters which are characteristic for spoiled bread could be used in the routine analysis of bread and indicate spoilage? Highlight it in manuscript.

Answer – I explain better – Usually the volatile compounds of food are important for the flavor, so in this case you are right they do not indicate the spoilage. In this case their increasing in the spoiled samples can only confirm the yeast activity. I eliminate the sentence.   

In the conclusion you mentioned gluten free bread. Results are missing for content of compounds of gluten free bread.

Answer – Thanks I correct GB not GFB.

Reviewer 2 Report

Comments and Suggestions for Authors

The authors of the manuscript "Cyberlindnera fabianii, an uncommon yeast responsible for gluten bread spoilage" in their research isolated a single strain of yeast responsible for the spoilage of local industrial gluten bread. The research was multivariate and the identification was supported by the analysis of alcohols, organic acids and esters, the concentration of which was lower in the uncontaminated samples. The experiment was properly planned and executed, but in order to present the results more fully, it requires some adjustments.

1.      The research material was poorly presented. The type of wheat flour needs to be supplemented, the value 0 is given in the manuscript, which is not known how to interpret.

2.      It was not specified how many hours after baking the bread was foiled, or at what temperature and how long it was stored.

3.      Clearly state what the sampling criterion was,

Author Response

Dear referee

Enclosed you will find a copy of our revised Manuscript ID: foods-3118685entitled “Cyberlindnera fabianii, an uncommon yeast responsible for gluten bread spoilage” submitted to Foods

The authors would like to thank the reviewers for their careful reading of the manuscript and the resulting constructive comments and suggestions. Basically, we agree with all the points raised by the reviewers, and wherever possible the manuscript has been modified as recommended. All reviewer comments are in black plain font, whereas our response is described in red plain font.

We have made the changes and corrections based on the reviewer’s suggestions. We evaluated the comments and prepared a point-by-point response to each one of them.

Reviewer 2

The authors of the manuscript "Cyberlindnera fabianii, an uncommon yeast responsible for gluten bread spoilage" in their research isolated a single strain of yeast responsible for the spoilage of local industrial gluten bread. The research was multivariate and the identification was supported by the analysis of alcohols, organic acids and esters, the concentration of which was lower in the uncontaminated samples. The experiment was properly planned and executed, but in order to present the results more fully, it requires some adjustments.

  1. The research material was poorly presented. The type of wheat flour needs to be supplemented, the value 0 is given in the manuscript, which is not known how to interpret.

Answer I add the composition – Lines 113 – 114 (Water 13%; Protein 11 %; Lipids 1%; carbohydrates 74%; Ash 0.65%) 

  1. It was not specified how many hours after baking the bread was foiled, or at what temperature and how long it was stored.

Answer Thanks – I add Lines 116-117 - The GB were folied after 6 hours from cooking and kept at room temperature. At day 15 the white spots were already visible.

  1. Clearly state what the sampling criterion was,

Answer Thanks – I explain well – Lines 124-125 -  All the spots of each spoiled packet were analyzed via the traditional method

Reviewer 3 Report

Comments and Suggestions for Authors

Dear authors, the manuscript "Cyberlindnera fabianii, an uncommon yeast responsible for gluten bread spoilage" is quite interesting and worth investigation. Please see some comments below:

1- Please double-check grammar and formatting;

2- Have you considered use other analytical approach instead SPME? I mean, it is easy, fast and so on, however, it is not acurate as liquid-liquid extraction, CG accoplated with head space. In addition, HPLC and fine analytical methods as proteomics can lead you a very deeper discussion.

3- The genetic identification seems quite aligned.

Regards

Author Response

Dear referee

Enclosed you will find a copy of our revised Manuscript ID: foods-3118685entitled “Cyberlindnera fabianii, an uncommon yeast responsible for gluten bread spoilage” submitted to Foods

The authors would like to thank the reviewers for their careful reading of the manuscript and the resulting constructive comments and suggestions. Basically, we agree with all the points raised by the reviewers, and wherever possible the manuscript has been modified as recommended. All reviewer comments are in black plain font, whereas our response is described in red plain font.

We have made the changes and corrections based on the reviewer’s suggestions. We evaluated the comments and prepared a point-by-point response to each one of them.

Reviwer 3

  • Please double-check grammar and formatting;

Answer Thanks – I did it

  • Have you considered use other analytical approach instead SPME? I mean, it is easy, fast and so on, however, it is not acurate as liquid-liquid extraction, CG accoplated with head space. In addition, HPLC and fine analytical methods as proteomics can lead you a very deeper discussion.

Answer – Thanks. Sorry, but I did not consider other methods – The SPME is well performed in my Department, so I used it. Probably you are right, but I never used it and this is the first time that a referees suggests me another method. Thanks 

  • The genetic identification seems quite aligned.

Answer – Thanks – This is the best possible alignment.

Round 2

Reviewer 1 Report

Comments and Suggestions for Authors

Dear Authors, 

Thank you very much for all answers and explanations, it is fine for me. 

Wish you all the best in the future work,